# XGAN: UNSUPERVISED IMAGE-TO-IMAGE TRANSLATION FOR MANY-TO-MANY MAPPINGS

## ABSTRACT

Style transfer usually refers to the task of applying color and texture information from a specific *style* image to a given *content* image while preserving the structure of the latter. Here we tackle the more generic problem of *semantic style transfer*: given two unpaired collections of images, we aim to learn a mapping between the corpus-level style of each collection, while preserving semantic content shared across the two domains. We introduce XGAN, a dual adversarial autoencoder, which captures a shared representation of the common domain semantic content in an unsupervised way, while jointly learning the domain-to-domain image translations in both directions. We exploit ideas from the domain adaptation literature and define a *semantic consistency loss* which encourages the model to preserve semantics in the learned embedding space. We report promising qualitative results for the task of face-to-cartoon translation. The cartoon dataset we collected for this purpose will also be released as a new benchmark for semantic style transfer.

## 1 INTRODUCTION

Image-to-image translation – learning to map images from one domain to another – covers several classical computer vision tasks such as style transfer (rendering an image in the style of a given input (Gatys et al., 2016)), colorization (mapping grayscale images to color images (Zhang et al., 2016)), super-resolution (increasing the resolution of an input image (Ledig et al., 2016)), or semantic segmentation (inferring pixelwise semantic labeling of a scene (Long et al., 2015)). In many cases, one can rely on supervision in the form of labels or paired samples. This assumption holds for instance for colorization, where ground-truth pairs are easily obtained by generating grayscale images from colored inputs.

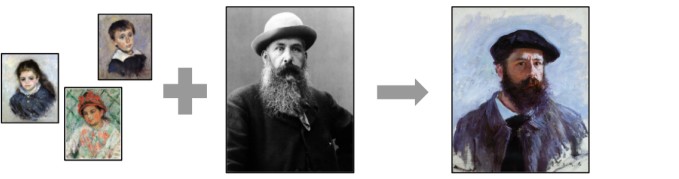 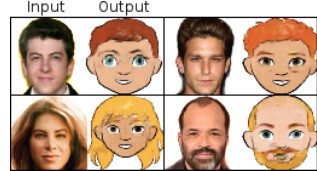

Figure 1: On the left, we depict a high-level motivational example for semantic style transfer, the task of adapting an image to the visual appearance of an other domain without altering its semantic content. The proposed XGAN applied on the face-to-cartoon task preserves important face semantics such as hair style or face shape (*right*).

In this work, we consider the task of *semantic style transfer*: learning to map an image from one domain into the style of another domain without altering its semantic content (see Figure 1). In that sense, our goal is akin to style transfer: We aim to transfer *style* while keeping *content* consistent. The key differences with traditional techniques are that (i) we work with image collections instead of having a single style image, and (ii) we aim to retain higher-level semantic content in the feature space rather than pixel-level structure. In particular, we experiment on the task of translating faces to cartoons while preserving their various facial attributes (hair color, eye color, etc.). Note that without loss of generality, a photo of a face can be mapped to many valid cartoons, and vice versa. Semantic style transfer is therefore a *many-to-many mapping* problem, for which obtaining labeled examples

is ambiguous and costly. Although this paper specifically focuses on the face-to-cartoon setting, many other examples fall under this category: mapping landscape pictures to paintings (where the different scene objects and their composition describe the input semantics), transforming sketches to images, or even cross-domain tasks such as generating images from text. In this setting, we only rely on two unlabeled training image collections or *corpora*, one for each domain, with no known image pairings across domains. Hence, we are faced with a double *domain shift*, first in terms of global domain appearance, and second in terms of the content distribution of the two collections.

Recent work (Kim et al., 2017; Zhu et al., 2017; Yi et al., 2017; Bousmalis et al., 2017) report good performance using GAN-based models for unsupervised image-to-image translation when the two input domains share similar pixel-level structure (e.g., horses and zebras) but fail for more general transformations (e.g., dogs and cats). Perhaps the best known recent example is CycleGAN (Zhu et al., 2017). Given two image domains $\mathcal{D}_1$ and $\mathcal{D}_2$, the model is trained with a pixel-level *cycle-consistency loss* which ensures that the mapping $g_{1 \to 2}$ from $\mathcal{D}_1$ to $\mathcal{D}_2$ followed by its inverse, $g_{2 \to 1}$, yields the identity function; i.e., $g_{1 \to 2} \circ g_{2 \to 1} = id$. However, we argue that such a pixel-level constraint is not sufficient in our case; the category of transformations we are interested in requires a constraint in *semantic space* even though the transformation occurs in the pixel space.

To this end, we propose XGAN ("Cross-GAN"), a dual adversarial autoencoder which learns a shared semantic representation of the two input domains in an unsupervised way, while jointly learning both domain-to-domain translations. In other words, the domain-to-domain translation $g_{1 \to 2}$ consists of an encoder $e_1$ taking inputs in $\mathcal{D}_1$, followed by a decoder $d_2$ with outputs in $\mathcal{D}_2$ (and likewise for $g_{2 \to 1}$) such that $e_1$ and $e_2$, as well as $d_1$ and $d_2$, are partially shared. The main novelty lies in how we constrain the shared embedding using techniques from the domain adaptation literature, as well as a novel *semantic consistency loss*. The latter ensures that the domain-to-domain translations preserve the semantic representation, i.e., that $e_1 \approx e_2 \circ g_{1 \to 2}$ and $e_2 \approx e_1 \circ g_{2 \to 1}$. Therefore, it acts as a form of self-supervision which alleviates the need for paired examples and preserves semantic feature-level information rather than pixel-level content. In the following section, we review relevant recent work before discussing the XGAN model in more detail in Section 3. In Section 4, we introduce CARTOONSET, our dataset of cartoon faces for research on semantic style transfer, which we are currently in the process of making publicly available. Finally, in Section 5 we report experimental results of XGAN on the face-to-cartoon task, and discuss various ablation experiments.

## 2  RELATED WORK

Recent literature suggests two main directions for tackling the semantic style transfer task: traditional style transfer and pixel-level domain adaptation. The first approach is inadequate as it only transfers texture information from a single style image, and therefore does not capture the style of an entire corpus. The latter category also fails in practice as it assumes pixel-level similarity which does not allow for significant structural change of the input. Instead, we draw inspiration from the domain adaptation and feature-level image-to-image translation literature.

**Style Transfer.**    Style transfer traditionally refers to the task of transferring the texture of a *specific* style image while preserving the pixel-level structure of an input content image (Gatys et al., 2016; Johnson et al., 2016). Recently, (Li & Wand, 2016; Liao et al., 2017) proposed to compare the style and generated image via a dense local patch-based matching approach in the feature space, as opposed to global feature matching, allowing for transformations between visually dissimilar domains. Still, these models only perform image-specific transfer rather than learning a global *corpus-level* style, and do not provide a meaningful joint semantic domain representation.

**Domain adaptation.**    XGAN relies on learning a shared semantic representation of both domains in an unsupervised setting. For this purpose, we make use of the domain-adversarial training scheme (Ganin et al., 2016). Moreover, recent domain adaptation work (Bousmalis et al., 2016; Shrivastava et al., 2017; Bousmalis et al., 2017) can be framed as semantic style transfer as they tackle the problem of mapping synthetic images, easy to generate, to natural images, which are more difficult to obtain. The generated samples are then used to train a model that can be applied to natural images. Contrary to our work however, they only consider pixel-level transformations.

**Image-to-Image translation.** Recent work (Kim et al., 2017; Zhu et al., 2017; Yi et al., 2017) tackle the unsupervised pixel-level image-to-image translation task by learning both cross-domain mappings jointly, each as a separate generative network, via a cycle-consistency loss which ensures that applying each mapping followed by its reverse yields the identity function. This intuitive form of self-supervision leads to good results for pixel-level transformations, but often fails to capture significant structural changes Zhu et al. (2017). In comparison, our proposed semantic consistency loss acts at the feature-level, allowing for more flexible transformations. Orthogonal to this work is UNIT (Liu et al., 2017). While also trained with pixel-level cycle-consistency, it consists of a coupled VAEGAN Larsen et al. (2015); Liu & Tuzel (2016) with a shared embedding bottleneck, similar to XGAN. However, UNIT assumes that sharing high-level layers in the architecture is sufficient to learn a joint representation of both domains, while XGAN's objective explicitly introduces the semantic consistency component.

The *Domain Transfer Network* (DTN) (Taigman et al., 2016; Wolf et al., 2017) is closest to our work. DTN is a single autoencoder trained to map images from a source to a target domain with self-supervised semantic-consistency feedback. It was also successfully applied to the problem of feature-level image-to-image translation, in particular to the face-to-cartoon problem. Contrary to XGAN however, the DTN encoder is pretrained and fixed, and is assumed to produce meaningful embeddings for both the face and the cartoon domains. This assumption is very restrictive, as off-the-shelf models pretrained on natural images do not necessarily generalize to other domains. In fact, while the reported results are convincing, we show in Section 5 that using a fixed encoder does not generalize well in the presence of large domain shift between the two input domains.

## 3 PROPOSED MODEL

Let $\mathcal{D}_1$ and $\mathcal{D}_2$ be two domains that differ in terms of *visual appearance* but share common *semantic content*. Note that while it is easier to think of domain semantics as a high-level notion, as for instance semantic attributes, we do not require such annotations in practice, but instead consider learning a feature-level representation that automatically captures these semantics without supervision. Our goal is thus to learn in an unsupervised fashion, i.e., without paired examples, a joint domain-invariant embedding that is semantically-consistent and meaningful for both domains; i.e., semantically similar inputs in both domains will be embedded nearby in the learned semantic space.

Architecture-wise, XGAN is a dual autoencoder on domains $\mathcal{D}_1$ and $\mathcal{D}_2$ (Figure 2(A)). We denote by $e_1$ the encoder and by $d_1$ the decoder for domain $\mathcal{D}_1$; likewise $e_2$ and $d_2$ for $\mathcal{D}_2$. For simplicity, we also denote by $g_{1\to2} = d_2 \circ e_1$ the transformation from $\mathcal{D}_1$ to $\mathcal{D}_2$; likewise $g_{2\to1}$ for $\mathcal{D}_2$ to $\mathcal{D}_1$.

The training objective can be decomposed into five main components:

the *reconstruction* loss, $\mathcal{L}_{rec}$, encourages the learned embedding to encode meaningful knowledge for each domain; the *domain-adversarial* loss, $\mathcal{L}_{dann}$, pushes embeddings from $\mathcal{D}_1$ and $\mathcal{D}_2$ to lie in the same subspace, bridging the domain gap at the semantic level; the *semantic consistency* loss, $\mathcal{L}_{sem}$, ensures that input semantics are preserved after domain translation; $\mathcal{L}_{gan}$ is a simple generative adversarial (GAN) objective, encouraging the model to generate more realistic samples, and finally, $\mathcal{L}_{teach}$ is an optional teacher loss that distills prior knowledge from a fixed pretrained teacher embedding, when available. The total loss function is defined as:

$$\mathcal{L}_{\text{XGAN}} = \mathcal{L}_{rec} + \omega_{dann}\mathcal{L}_{dann} + \omega_{sem}\mathcal{L}_{sem} + \omega_{gan}\mathcal{L}_{gan} + \omega_{teach}\mathcal{L}_{teach}, \qquad (1)$$

where the $\omega$ hyper-parameters control the contributions from each of the individual objectives. An overview of the model is given in Figure 2, and we discuss each objective in more detail in the rest of this section.

**Reconstruction loss.** $\mathcal{L}_{rec}$ encourages the model to encode enough information on each domain for the input to be reconstructed by the autoencoder. More specifically $\mathcal{L}_{rec} = \mathcal{L}_{rec,1} + \mathcal{L}_{rec,2}$ is the sum of reconstruction losses for each domain.

$$\mathcal{L}_{rec,1} = \mathbb{E}_{\mathbf{x}\sim p_{\mathcal{D}_1}} \left( \|\mathbf{x} - d_1(e_1(\mathbf{x}))\|_2 \right), \text{ and likewise for } \mathcal{L}_{rec,2} \qquad (2)$$

**Domain-adversarial loss.** $\mathcal{L}_{dann}$ is the domain-adversarial loss between $\mathcal{D}_1$ and $\mathcal{D}_2$, as introduced in Ganin et al. (2016). It encourages the embeddings learned by $e_1$ and $e_2$ to lie in the same

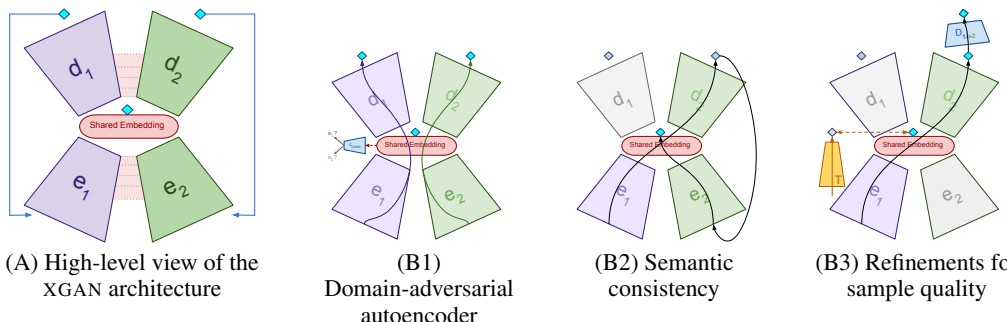

| (A) High-level view of the XGAN architecture | (B1) Domain-adversarial autoencoder | (B2) Semantic consistency | (B3) Refinements for sample quality |

Figure 2: The XGAN architecture (A) is trained via an objective that encourages the model to learn a meaningful joint embedding (B1) ($\mathcal{L}_{rec}$ and $\mathcal{L}_{dann}$), which should be preserved through domain translation (B2) ($\mathcal{L}_{sem}$), while producing output samples of good quality (B3) ($\mathcal{L}_{gan}$ and $\mathcal{L}_{teach}$)

subspace. In particular, it guarantees the soundness of the cross-domain transformations $g_{1\rightarrow2}$ and $g_{2\rightarrow1}$. More formally, this is achieved by training a binary classifier, $c_{dann}$, on top of the embedding layer to categorize encoded images from *both* domains as coming from either $\mathcal{D}_1$ or $\mathcal{D}_2$ (see Figure 2(B1)). $c_{dann}$ is trained to maximize its classification accuracy $\mathcal{L}_{dann}$ while the encoders $e_1$ and $e_2$ simultaneously strive to minimize it, i.e., to confuse the domain-adversarial classifier. Denoting model parameters by $\theta$ and a classification loss function by $\ell$ (e.g., cross-entropy), we have:

$$\min_{\theta_{e_1},\theta_{e_2}} \max_{\theta_{dann}} \mathcal{L}_{dann}, \text{ where } \mathcal{L}_{dann} = \mathbb{E}_{p_{\mathcal{D}_1}} \ell(1, c_{dann}(e_1(\mathbf{x}))) + \mathbb{E}_{p_{\mathcal{D}_2}} \ell(2, c_{dann}(e_2(\mathbf{x}))) \quad (3)$$

**Semantic consistency loss, $\mathcal{L}_{sem}$.** Our key contribution is a semantic consistency feedback loop that acts as self-supervision for the cross-domain translations $g_{1\rightarrow2}$ and $g_{2\rightarrow1}$. It reinforces the action of the domain-adversarial loss $\mathcal{L}_{dann}$ by mapping the embedding of an input image and the embedding of its translated counterpart to the same point. Intuitively, we want the semantics of input $\mathbf{x} \in \mathcal{D}_1$ to be preserved when translated to the other domain, $g_{1\rightarrow2}(\mathbf{x}) \in \mathcal{D}_2$, and similarly for the reverse mapping. However this consistency property is hard to assess at the pixel-level as we do not have paired data and pixel-level metrics are suboptimal for image comparison. Instead, we introduce a feature-level semantic consistency loss, which encourages the network to preserve the learned embedding during domain translation. Formally, $\mathcal{L}_{sem} = \mathcal{L}_{sem,1\rightarrow2} + \mathcal{L}_{sem,2\rightarrow1}$, where:

$$\mathcal{L}_{sem,1\rightarrow2} = \mathbb{E}_{\mathbf{x}\sim p_{\mathcal{D}_1}} \|e_1(\mathbf{x}) - e_2(g_{1\rightarrow2}(\mathbf{x}))\|, \text{ where } \|\cdot\| \text{ is a distance between vectors.} \quad (4)$$

$\mathcal{L}_{sem,2\rightarrow1}$ is defined in the same way for the transformation from $\mathcal{D}_2$ to $\mathcal{D}_1$.

**GAN objective, $\mathcal{L}_{gan}$.** Although the key aim of XGAN is to learn a joint meaningful and semantically consistent embedding, we find that generating realistic image transformations has a crucial positive effect as the produced samples are fed back through the encoders when computing the semantic consistency loss: Making the transformed distribution $p(g_{2\rightarrow1}(\mathcal{D}_2))$ as close as possible to the original domain $p(\mathcal{D}_1)$ ensures that the encoder $e_1$ does not have to cope with an additional domain shift. Therefore, with the purpose of improving sample quality, we define $\mathcal{L}_{gan} = \mathcal{L}_{gan,1\rightarrow2} + \mathcal{L}_{gan,2\rightarrow1}$, where $\mathcal{L}_{gan,1\rightarrow2}$ is a state-of-the-art GAN objective (Goodfellow et al., 2014) where the generator $g_{1\rightarrow2}$ is paired against the discriminator $D_{1\rightarrow2}$ (and likewise for $g_{2\rightarrow1}$ and $D_{2\rightarrow1}$). The models are trained jointly in an adversarial scheme where $D_{1\rightarrow2}$ strives to distinguish generated samples from real samples in $\mathcal{D}_2$, while the generator aims to produce samples that confuse the discriminator, i.e.,

$$\min_{\theta_{g_{1\rightarrow2}}} \max_{\theta_{D_{1\rightarrow2}}} \mathcal{L}_{gan,1\rightarrow2}, \text{ where} \quad (5)$$

$$\mathcal{L}_{gan,1\rightarrow2} = \mathbb{E}_{\mathbf{x}\sim p_{\mathcal{D}_2}} (\log(D_{1\rightarrow2}(\mathbf{x}))) + \mathbb{E}_{\mathbf{x}\sim p_{\mathcal{D}_1}} (\log(1 - D_{1\rightarrow2}(g_{1\rightarrow2}(\mathbf{x})))) \quad (6)$$

Once again $\mathcal{L}_{gan,2\rightarrow1}$ is the symmetric version for the transformation from $\mathcal{D}_2$ to $\mathcal{D}_1$.

**Teacher loss, $\mathcal{L}_{teach}$.** We introduce an optional component to easily incorporate prior knowledge in the model when available, i.e., when working in a semi-supervised framework. $\mathcal{L}_{teach}$ encourages the learned embeddings to lie in a region of the subspace defined by the output of the representation

layer of a teacher network $T$. In other words, it distills knowledge from a pretrained teacher and constrains the embeddings to a more meaningful subregion (relative to the task on which $T$ was trained), which can be seen as a form of regularization of the learned embedding. $\mathcal{L}_{teach}$ is asymmetric by definition. It should not be used for both domains simultaneously as each term would potentially push the learned embedding in two different directions. Assuming it is applied to domain $\mathcal{D}_1$, leads to the following definition:

$$\mathcal{L}_{teach} = \mathbb{E}_{\mathbf{x} \sim p_{\mathcal{D}_1}} \| T(\mathbf{x}) - e_1(\mathbf{x}) \|, \text{ where } \| \cdot \| \text{ is a distance between vectors.}$$

### 3.1 ARCHITECTURE AND TRAINING PROCEDURE

We use a simple mirrored convolutional architecture for the autoencoder. It consists of 5 convolutional blocks for each encoder, the two last ones being shared across domains, and likewise for the decoders (5 deconvolutional blocks with the two first ones shared). This encourages the model to learn shared representations at different levels of the architecture rather than only in the middle layer. For the teacher network, we use the highest convolutional layer of FaceNet (Schroff et al., 2015), a state-of-the-art model pretrained for the task of face recognition. Note that FaceNet was trained on natural images only, i.e., it does not contain any prior knowledge of the cartoon domain. A more detailed description is given in Appendix 7.1.

The XGAN training objective is obtained by minimizing Equation (1). In particular, the two adversarial losses ($\mathcal{L}_{gan}$ and $\mathcal{L}_{dann}$) leads to minmax optimization problems that require careful optimization. For the GAN loss $\mathcal{L}_{gan}$, we use a standard adversarial training scheme Goodfellow et al. (2014). Note that in order to ease training, we only use one of the discriminators in practice, namely $D_{1 \rightarrow 2}$ which corresponds to the face-to-cartoon path, our target application. We first update the parameters of the generators $g_{1 \rightarrow 2}$ and $g_{2 \rightarrow 1}$ in one step. We then keep these fixed and update the parameters for the discriminator $D_{1 \rightarrow 2}$. Finally, we train the model by iterating this alternating process. The adversarial training scheme for $\mathcal{L}_{dann}$ can be easily implemented in practice by connecting the classifier $c_{dann}$ and the embedding layer *via* a gradient reversal layer (Ganin et al., 2016): The feed-forward pass is unaffected, however the gradient is backpropagated to the encoders with a sign-inversion representing the minmax alternation. This update is performed in the same step as for the generator parameters. Finally, we use ADAM optimization (Kingma & Ba, 2015) with an initial learning rate of 0.0001 to train the model.

## 4 THE CARTOONSET DATASET[1]

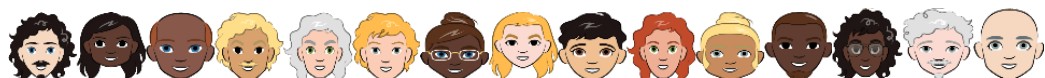

Figure 3: Random samples from our cartoon dataset, CartoonSet. Each cartoon face is composed of 16 discrete attributes resulting in the order of 100 million possible cartoon faces.

Although previous work has tackled the task of transforming frontal faces to a specific cartoon style, there is currently no such dataset publicly available. For this purpose, we introduce a new dataset, CartoonSet, which we will release publicly to further aid research on this topic.

Each cartoon face is composed of 16 components including 12 facial attributes (e.g., facial hair, eye shape, etc) and 4 color attributes (such as skin or hair color) which are chosen from a discrete set of RGB values. The number of options per attribute category ranges from 3, for short/medium/long chin length, to 111, for the largest category, hairstyle. Each of these components and their variation were drawn by the same artist, resulting in approximately 250 cartoon components artworks and $10^8$ possible combinations. Furthermore, the artwork components are divided into a fixed set of layers that define a Z-ordering for rendering. For instance, face shape is defined on a layer below eyes and glasses, so that the artworks are rendered in the correct order. Hair style is a more complex case and needs to be defined on two layers, one behind the face and one in front. There are 8 total layers: hair back, face, hair front, eyes, eyebrows, mouth, facial hair, and glasses. The mapping from attribute to artwork is also defined by the artist such that any random selection of attributes produces a visually

---

[1]We are currently in the process of releasing the dataset.

appealing cartoon without any misaligned artwork; this sometimes involves handling interaction between attributes. For example, the proper way to display a "short beard" changes for different face shapes, which required the artist to create a "short beard" artwork for each face shape.

We create the CartoonSet dataset from arbitrary cartoon faces by randomly sampling a value for each attribute. The corresponding artworks are rendered back-to-front. We then filter out unusual hair colors (pink, green etc) or unrealistic attribute combinations, which results in a final dataset of approximately $9,000$ cartoons. In particular, the filtering step guarantees that the dataset only contains realistic cartoons, while being completely unrelated to the source dataset.

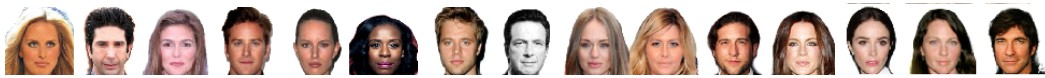

Figure 4: Random samples from the centered aligned VGG-Face dataset.

## 5 EXPERIMENTS

We experimentally evaluate our XGAN model on *semantic style transfer*; more specifically, on the task of converting images of frontal faces (source domain) to images of cartoon avatars (target domain) given an unpaired collection of such samples in each domain. Our source domain is composed of real-world frontal-face images from the VGG-Face dataset (Parkhi et al., 2015). In particular, we use an image collection consisting of 18,054 uncropped celebrity frontal face pictures. As a pre-processing step, we align the faces based on eyes and mouth location and remove the background. The target domain is the cartoon style we introduced in Section 4. The corresponding training image collection consists of 9,000 cartoon images that we center-align by localizing the center of the irises, the center of the mouth, and tip of the nose. Finally, we randomly select and take out 20% of the images from each dataset for testing purposes, and use the remaining 80% for training. For our experiments we also resize all images to $64 \times 64$. As shown in Figures 3 and 4, the two domains vary significantly in appearance. In particular, cartoon faces are rather simplistic compared to real faces, and do not display as much variety (e.g., noses or eyebrows only have a few shape options). Furthermore, we observe a major content distribution shift between the two domains due to the way we collected the data: for instance, certain hair color shades (e.g., bright red, gray) are over-represented in the cartoon domain compared to real faces. Similarly, the cartoon dataset contains many samples with eyeglasses while the source dataset only has a few.

**Baseline comparison.** Our primary evaluation result is a qualitative comparison between the Domain Transfer Network (DTN) (Taigman et al., 2016) and XGAN on the semantic style transfer problem outlined above. To the best of our knowledge, DTN is the current state of the art for semantic style transfer given unpaired image corpora from two domains with significant visual shift. In particular, DTN was also applied to the task of transferring face pictures to cartoons (bitmojis) in the original paper[2]. See Section 2 for a more detailed introduction. Figure 5 shows the performance of both DTN and XGAN applied to random VGG-Face samples from the test set to produce cartoon versions of each sample. For both models, we present random samples produced with the best set of hyperparameters we found. Evaluation metrics for style transfer are still an active research topic with no good solution yet. Hence we choose optimal hyperparameters by manually evaluating the quality of resulting samples, focusing on accurate transfer of semantic attributes, similarity of the resulting sample to the target domain, and crispness of samples.

It is clear from Figure 5 that DTN fails to capture the transformation function that semantically stylizes frontal faces to cartoons from our target domain. In contrast, XGAN is able to produce sensible cartoons both in terms of the style domain – the resulting cartoons look crisp and respect the specific CartoonSet style – and in terms of semantic similarity to the input samples from VGG-Face. There are some failure cases such as hair or skin color mismatch, which emerge from the weakly supervised nature of the task and the significant content shift between the two domains (e.g., red hair is over-represented in the target cartoon dataset). We also report selected XGAN samples

---

[2]The original DTN code and dataset is not publicly available, hence we instead report results from our implementation applied to the VGG-Face to CartoonSet setting.

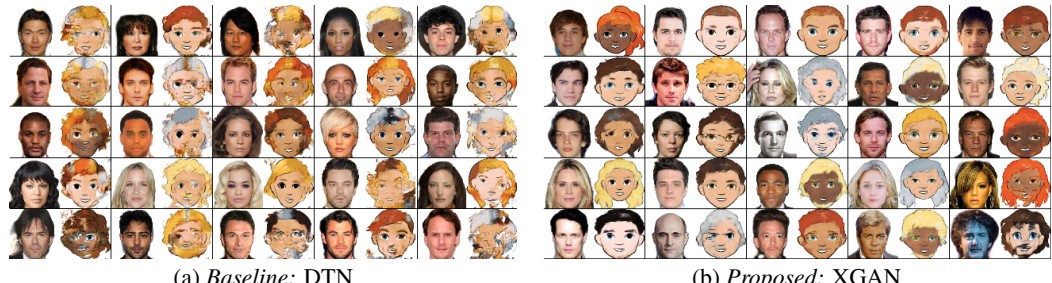

(a) *Baseline:* DTN          (b) *Proposed:* XGAN

Figure 5: A qualitative comparison between DTN and XGAN. In both cases we present random test samples for the face-to-cartoon transformation with optimal hyperparameters. The tables are prganized row-wise where each face input is mapped to the cartoon face immediately on its right.

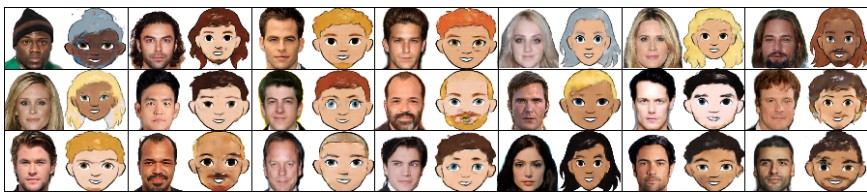

Figure 6: Selected samples generated by XGAN on the VGG-Face to CartoonSet task.

that we think best illustrate its semantic consistency abilities in Figure 6. Finally, additional random samples for both cross-domain mappings are available in Appendix 7.3.

We believe the failure of DTN is primarily due to its assumption of a fixed joint encoder for both domains. Although the decoder learns to reconstruct inputs from the target domain almost perfectly, the semantics are not well preserved across domains and the decoder yields samples of poor quality for the domain transfer. In fact, FaceNet was originally trained on real faces inputs, hence there is no guarantee it can produce a meaningful representation for CartoonSet samples. In contrast to our dataset, the target bitmoji domain in (Taigman et al., 2016) is visually closer to real faces, as bitmojis are more realistic and customizable than the cartoon style domain we introduce here. This might explain the good reported performance even with a fixed encoder. Our experiments suggest that using a fixed encoder is a very restrictive assumption that does not adapt well to new scenarios. We also report results from a finetuned DTN in Appendix 7.2 and 7.3, which yields samples of better quality than the original DTN. However, this setup is very sensitive to training hyperparameters and prone to mode collapse.

**Ablation study.** We conduct a number of insightful ablation experiments on XGAN. We first consider training only with the reconstruction loss $\mathcal{L}_{rec}$ and domain-adversarial loss $\mathcal{L}_{dann}$. In fact these form the core domain adaptation component in XGAN and, as we will show, are already able to capture basic semantic knowledge across domains in practice. Secondly we experiment with the semantic consistency loss and teacher loss. We show that both have a constraining effect on the embedding space which contributes to improving the sample consistency. We also show in Appendix 7.4.1 that the GAN loss, even though it makes training more complex, is necessary for producing samples of good quality and cannot be replaced with simpler image smoothness objectives.

We first experiment on XGAN with only the reconstruction and domain-adversarial losses active. This component prompts the model to (i) encode enough information for each decoder to correctly reconstruct images from the corresponding domain and (ii) to ensure that the embedding lies in a common subspace for both domains. In practice in this setting, the model is robust to hyperparameter choice and does not require much tuning to converge to a good regime, i.e., low reconstruction error and around 50% accuracy for the domain-adversarial classifier. As a result of (ii), applying each decoder to the output of the other domain's encoder yields reasonable cross-domain translations, albeit of low quality (see Figure 7). Furthermore, we observe that some simple semantics such as skin tone or gender are overall well preserved by the learned embedding due to the shared

autoencoder structure. For comparison, failure modes occur in extreme cases, e.g., when the model capacity is too small, in which case transferred samples are of poor quality, or when $\omega_{dann}$ is too low. In the latter case, the source and target embeddings are easily distinguishable and the cross-domain translations do not look realistic (see Appendix 7.4).

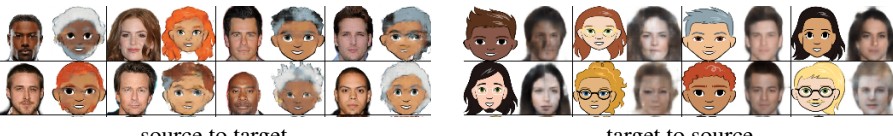

source to target                 target to source

Figure 7: Test results for XGAN with the reconstruction and domain-adversarial losses only

Secondly, we investigate the benefits of adding semantic consistency in XGAN via the following three components: *Sharing high-level layers* in the autoencoder leads the model to capture common semantics earlier in the architecture. In general, high-level layers in convolutional neural networks are known to encode semantic information. We perform a few experiments when sharing only the middle layer in the dual autoencoder. As expected, the resulting embedding does not capture relevant shared domain semantics. Second, we use the *semantic consistency loss* as self-supervision for the learned embedding, ensuring that it is preserved through the cross-domain transformations. It also reinforces the action of the domain-adversarial loss as it constrains embeddings from the two input domains to lie close to each other. Finally, the optional *teacher loss* leads the learned source embedding to lie near the teacher output (in our case, FaceNet's representation layer), which is meaningful for real faces. It acts in conjunction which the domain-adversarial loss and semantic consistency loss which bring the source and target embedding distributions closer to each other.

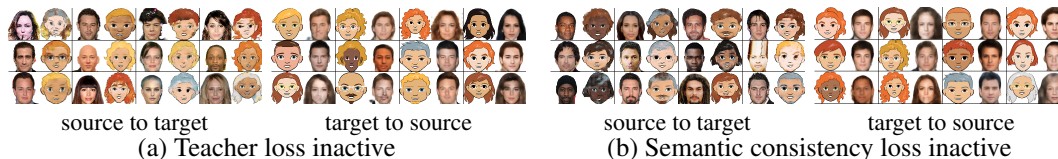

source to target       target to source         source to target       target to source
(a) Teacher loss inactive           (b) Semantic consistency loss inactive

Figure 8: Results of ablating the teacher loss (left) and semantic consistency loss (right) in XGAN.

In Figure 8 we report random test samples for both domain-to-domain translations when ablating the teacher loss and semantic consistency loss respectively. While it is hard to draw conclusions from qualitative results, it seems that the teacher network has a positive regularization effect on the learned embedding by guiding it to a more reasonable region of the space: Training the model without the teacher loss (Figure 8(a)) yields more distorted samples, especially when the input is an outlier, e.g., person wearing a hat, or cartoons with unusual hairstyles (Figure 5(b)). Conversely, when the semantic consistency is inactive (Figure 8(b)), the generated samples overall display less variety. In particular, rare attributes (e.g., unusual hairstyle) are not as well preserved as when the semantic consistency loss is present.

**Discussions and Limitations.** Our initial motivation for XGAN was to tackle the *semantic style transfer* problem in a fully unsupervised framework by combining techniques from domain adaptation and image-to-image translation. We first observe that using a simple setup where a partially shared dual autoencoder is trained with reconstruction losses and a domain-adversarial loss already suffices to produce an embedding that captures basic semantics rather well (for instance, skin tone). However, the generated samples are of poor quality and fine-grained attributes such as facial hair are not well captured. These two problems are greatly diminished after adding the GAN loss and the proposed semantic consistency loss, respectively. Failure cases still exist, especially on non-representative input samples (e.g., a person wearing a hat) which are mapped to unrealistic cartoons. Adding the teacher loss reduces this problem by regularizing the learned embedding, however it requires additional supervision and makes the model dependent on the specific representation provided by the teacher network. Future work will focus on evaluating XGAN on more tasks. In particular, , while we introduced XGAN as a solution to semantic style transfer, we think the model goes be-

yond this scenario and could be applied to classical domain adaptation problems, where quantitative evaluation becomes possible.

## 6 CONCLUSIONS

In this work, we introduced XGAN, a model for unsupervised domain translation applied to the task of semantically-consistent style transfer. In particular, we argue that learning image-to-image translation between two structurally different domains requires passing through a high-level joint semantic representation while discarding local pixel-level dependencies. Additionally, we proposed a semantic consistency loss acting on both domain translations as a form of self-supervision.

We reported promising experimental results on the task of mapping the domain of face images to cartoon avatars that clearly outperform the current baseline. We also showed that additional weak supervision, such as a pretrained feature representation, can easily be added to the model in the form of teacher knowledge. While not necessary, it acts as a good regularizer for the learned embeddings and generated samples. This can be particularly useful for natural image data as off-the-shelf pretrained models are abundant.

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

# 7 APPENDIX

## 7.1 ARCHITECTURE DETAILS

**Autoencoder.** Encoders take 64x64 images as input, which are then fed through five 2D convolutional blocks. Two fully-connected layers are applied to the last feature map in order to obtain the embedding vector. Finally, we normalize the embedding vector so that it lies in the unit ball. We use the cosine distance for all embedding comparisons (for the semantic consistency and teacher loss). The architecture for the decoder is a mirrored version of the encoder. From the initial flat embedding layer, we apply a sequence of five deconvolutions, the last block outputting an 64x64 color image. For both the encoder and decoder, the two highest-level (de)convolutional blocks are shared across domains. This encourages the model to learn shared representations at different levels of the architecture rather than only in the middle layer. A detailed overview of the architecture is presented in Appendix 7.1.

**Discriminator.** The discriminator architecture is very similar to the encoder architecture with the difference that it only needs to output one logit for each input image, representing its binary classification decision. In practice, we use a smaller architecture for the discriminator as it often tends to be too powerful and easily distinguish between real and transformed images.

| Layer | Size |
|---|---|
| Inputs | 64x64x3 |
| conv1 | 32x32x32 |
| conv2 | 16x16x64 |
| (//) conv3 | 8x8x128 |
| (//) conv4 | 4x4x256 |
| (//) FC1 | 1x1x1024 |
| (//) FC2 | 1x1x1024 |
| $L_2$ norm. | 1x1x1024 |

(a) Encoder architecture

| Layer | Size |
|---|---|
| Inputs | 1x1x1024 |
| (//) deconv1 | 4x4x512 |
| (//) deconv2 | 8x8x256 |
| deconv3 | 16x16x128 |
| deconv4 | 32x32x64 |
| deconv5 | 64x64x3 |

(b) Decoder architecture

| Layer | Size |
|---|---|
| Inputs | 64x64x3 |
| conv1 | 32x32x16 |
| conv2 | 16x16x32 |
| conv3 | 8x8x32 |
| conv4 | 4x4x32 |
| FC1 | 1x1x1 |

(c) Discriminator architecture

Table 1: Overview of the XGAN architecture used in practice. The encoder and decoder have the same architecture for both domains, and (//) indicates that the layer is shared across domain.

We also report details of the XGAN architecture in Table 1. Note that all layers except the last ones are followed by batch normalization. We also use ReLU as activation function for each of them, except for the last deconvolution of the decoders which uses hyperbolic tangent activation function.

## 7.2 FINETUNING THE DTN ENCODER

As we noted when experimenting with the DTN, its main drawback seems to come from the assumption to keep a fixed pretrained encoder in the model. Following this observation, we perform

another experiment in which we finetune the FaceNet encoder relatively to the semantic consistency loss, additionally to the decoder parameters.

While this yields visually better samples (see Figure 9(b)), it also raises the classical domain adaptation issue of guaranteeing that the initial FaceNet embedding knowledge is preserved when retraining the embedding. For comparison, XGAN exploits a teacher network that can be used to distill prior domain knowledge throughout training, when available. Secondly, this finetuned DTN is prone to mode collapse. In fact, the encoder is now only trained relatively to the semantic consistency loss which can be easily minizized by mapping each domain to the same point in the embedding space, leading to the same cartoon being generated for all of them. In XGAN, the source embeddings are regularized by the reconstruction loss on the source domain. This allows us to learn a joint domain embedding from scratch in a proper domain adaptation framework.

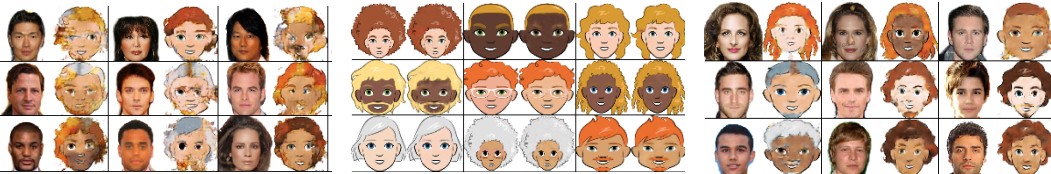

(a) Random generated samples (left) and reconstructions (right) with fixed FaceNet embedding

(b) Random generated samples with a fine-tuned FaceNet encoder

Figure 9: Reproducing the Domain Transfer Network performs badly in our experimental setting (a); fine-tuning the encoder yields better results (b) but is unstable for training in practice.

## 7.3 EXTENSIVE QUALITATIVE EVALUATION

As mentioned in the main text, the DTN baseline fails to capture a meaningful shared embedding for the two input domains. Instead, we consider and experiment with three different models to tackle the semantic style transfer problem. Selected samples are reported in Figure 10:

- **Finetuned DTN**, as introduced previously. In practice, this model yields satisfactory samples but is very sensitive to hyperparameter choice and often collapses to one model.
- **XGAN with $\mathcal{L}_{rec}$ and $\mathcal{L}_{dann}$ active only** corresponds to a simple domain-adaptation setting: the proposed XGAN model where only the reconstruction loss $\mathcal{L}_{rec}$ and the domain-adversarial loss $\mathcal{L}_{dann}$ are active. We observe that semantics are globally well preserved across domains although the model still makes some basic mistakes (e.g., gender misclassifications) and the samples quality is poor.
- **XGAN**, the full proposed model, yields the best visual samples out of the models we experiment on. In the rest of this section, we report a detailed study on its different components and possible failure modes.

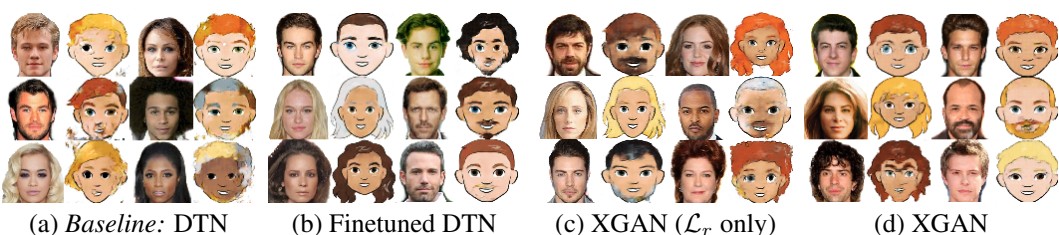

(a) *Baseline:* DTN     (b) Finetuned DTN     (c) XGAN ($\mathcal{L}_r$ only)     (d) XGAN

Figure 10: Cherry-picked samples for the DTN baseline and three improved models we consider for the semantic style transfer task

In Figure 11 we also report a more extensive random selection of samples produced by XGAN. Note that we only used a discriminator for the source to target path (i.e., $\mathcal{L}_{gan,2\to1}$ is inactive); in fact the GAN objective tends to make training more unstable so we only use one for the transformation we care most about for this specific application, i.e., faces to cartoons. Other than the GAN objective, the model appears to be robust to the choice of hyperparameters.

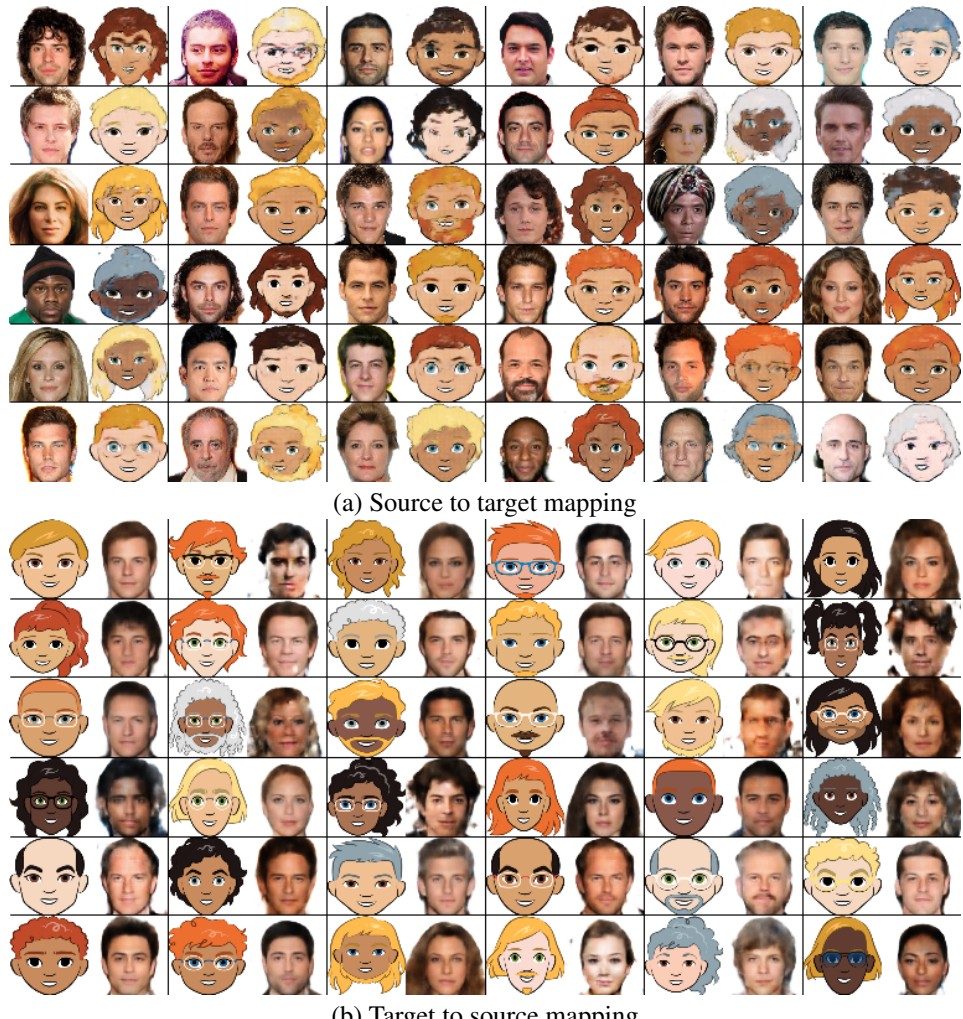

(a) Source to target mapping

(b) Target to source mapping

Figure 11: Random samples obtain when applying a trained XGAN on faces from the testing set

Overall, the cartoon samples are visually very close to the original dataset and main identity characteristics such as face shape, hair style, skin tone, etc., are well preserved between the two domains. The main failure mode appears to be mismatched hair color: in particular, bright red hair appear very often in generated samples which is likely due to its abundance in the training cartoon dataset. In fact, when looking at the target to source generated samples, we observe that this color shade often gets mapped to dark brown hair in the real face domain. One could expect the teacher network to regularize the hair color mapping, however FaceNet was originally trained for face identification, hence is most likely more sensitive to structural characteristics such as face shape. More generally, most mistakes are due to the shift in *content* distribution rather than *style* distribution between the two domains. Other examples include bald faces being mapped to cartoons with light hair (most likely due to the lack of bald cartoon faces and the model mistaking the white background for hair color). Also, eyeglasses on cartoon faces disappear when mapped to the real face domain (only very few faces in the source dataset wear glasses).

## 7.4 FAILURE MODE WHEN TRAINING WITH $\mathcal{L}_{rec}$ AND $\mathcal{L}_{dann}$

In Figure 12 we report examples of failure cases when $\omega_{dann}$ is too high in the setting with the reconstruction and domain-adversarial loss only: The domain-adversarial classifier $c_{dann}$ reaches perfect accuracy and cross-domain translation fails.

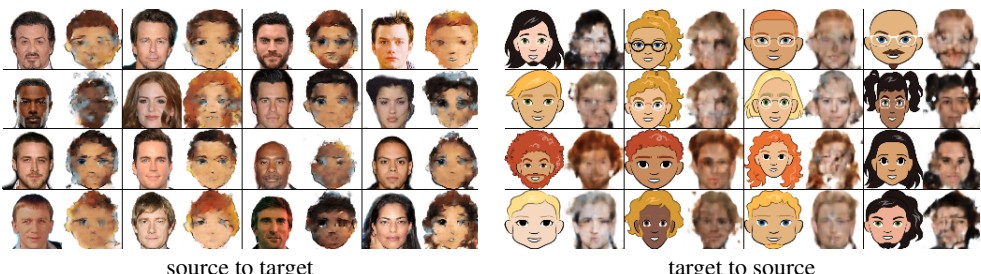

source to target                                            target to source

Figure 12: Random test samples for both cross-domain translations in the failure mode for the $\mathcal{L}_{rec} + \mathcal{L}_{dann}$ only XGAN setting

### 7.4.1 GAN LOSS ABLATION EXPERIMENT

As mentioned Section 3.1, we only use a GAN loss term for the source $\rightarrow$ target translation, to ease training. This prompts the face-to-cartoon path to generate more realistic samples. As expected, when the GAN loss is inactive, the generated samples are noisy and unrealistic (see Figure 13(a)). For comparison, tackling the low quality problem with simpler regularization techniques such as using total variation smoothness loss leads to more uniform samples but significantly worsen their blurriness on the long term (see Figure 13(b)). This shows the importance of the GAN objective for image generation applications, even though it makes the training process more complex.

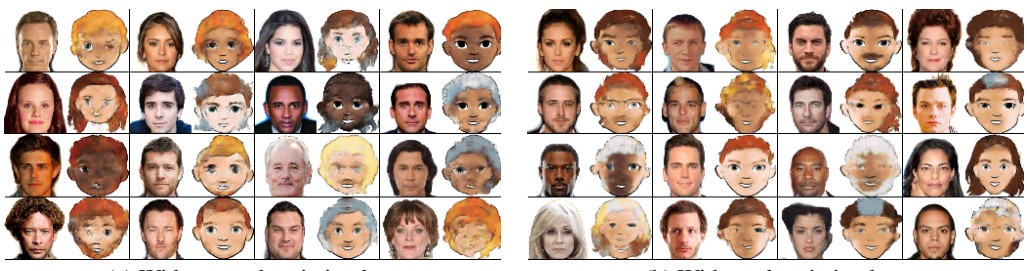

(a) Without total variation loss                    (b) With total variation loss

Figure 13: Test samples for XGAN when the GAN loss $\mathcal{L}_{ga}$ is inactive

