# OpenReview forum: "XGAN: Unsupervised Image-to-Image Translation for many-to-many Mappings"
_ICLR.cc/2018/Conference — Reject_

### Official Review · AnonReviewer3 · 2017-11-26
**Due to limited novelty, lack of clarity in presentation, and poor experimental validation, the reviewer recommends rejecting the paper.**

**Rating:** 3
**Confidence:** 5

**Review:**



- Lack of novelty

The paper has very limited novelty since the proposed method is a straightforward combination of two prior works on the same topic (unpair/unsupervised image translation or cross-domain image generation) where the two prior works are the DTN work [a] and the UNIT [b] work. To be more precise, the proposed method utilizes the weight-sharing design for enforcing the shared latent space constraint proposed in the UNIT work [b] and the feature consistency loss term for ensuring common embedding in the DTN work [a] for solving the ill-posed unpaired/unsupervised image-to-image translation problem. Since the ideas are already published in the prior work, the paper does not contribute additional knowledge to the problem.

In addition, the combination is done in a careless manner. First of all, the paper proposes jointly minimizing the common embedding loss [a] and the domain adversarial loss [b]. However, minimizing the common embedding loss [a] also results in minimizing the domain adversarial loss [c]. This can be easily seen as when the embeddings are the same, no discriminators can tell them apart. This suggests that the paper fails to see the connection and blindly put the two things together. Moreover, given the generators, minimizing the common embedding loss also results in minimizing the cycle-consistency loss [d]. As the UNIT work [b] utilize both the weight-sharing constraint and cycle-consistency loss, the proposed method becomes a close variant to the UNIT work [b].

- Poor experimental verification

The paper only shows visualization results on translating frontal face images to cartoon images in the resolution of 64x64. This is apparently short as compared to the experimental validations done in several prior works [a,b,d]. In the CycleGAN work [d], the results are shown on several translation tasks (picture to painting, horse to zebra, map to image, and different scenarios) in a resolution of 256x256. In the UNIT work [b], the results are shown in various street scene (sunny to rainy, day to night, winter to summer, synthetic to real) and animal portraits (cat species and dog breeds) where the resolution is up to 640x480. In the DTN [a] and UNIT [b] work, promising domain adaptation results (SVHN to MNIST) are reported. Due to the shortage of results, the credibility of the paper is damaged.

- Lack of clarity in presentation

The paper tends to introduces new key words for existing one. For example, the "semantic style transfer" is exactly the unpaired/unsupervised image-to-image translation or cross-domain image generation. It is not clear why the paper needs to introduce the new keyword. Also, the Coupled GAN work [e] is the first work that utilizes both weight-sharing (shared latent space assumption) and GAN for unpaired/unsupervised image-to-image translation. It is unfortunately that the paper fails to refer to this closely related prior work.

[a] Yaniv Taigman, Adam Polyak, Lior Wolf "Unsupervised Cross-Domain Image Generation", ICLR 2017

[b] Ming-Yu Liu, Thomas Breuel, Jan Kautz "Unsupervised Image-to-Image Translation Networks", NIPS 2017

[c] YaroslavGanin et al. "Domain-adversarial Training of Neural Networks" JMLR 2016

[d] Jun-Yan Zhu, Taesung Park, Philip Isola, and Alexei A. Efros "Unpaired Image-to-Image Translation Using Cycle-consistent Adversarial Networks" ICCV 2017

[e] Ming-Yu Liu, Oncel Tuzle "Coupled Generative Adversarial Networks", NIPS 2016

---

> ### Author Response · Authors · 2018-01-05
> **Response to reviews**
>
> We thank the reviewer for their detailed comment. We take note that the experiments section and comparison to baselines are lacking. In the following, we propose some clarifications to the specific issues raised in the review:
>
> Semantic style transfer
> -------------------------------
> We introduce the semantic style transfer keyword as a distinction to pixel-level translation tasks. While the notion of "unsupervised image-to-image translation" should be general enough, in recent work, this task often refers to pixel-level translation, where the input and output images have very similar structure (e.g., as you mentioned: horses to zebras, sunny to rain). Instead, we focused on translation tasks allowing for significant structure changes while retaining semantic content, such as the face-to-cartoon task introduced in DTN.
>
> Loss redundancy
> -----------------------
> The semantic consistency [a] and domain-adversarial loss [b] are indeed redundant (in the sense that low value of [a] implies low value of [b]) when the model perfectly maps inputs to the correct target domain; however this is not necessarily the case in practice, e.g. at the beginning of training.
> More specifically, the domain-adversarial [b] loss makes embeddings from D1 and D2 lie in close subspaces, while the semantic consistency loss makes embedding e1(x) close to e2 o d2 o e1(x) (and vice-versa) for a specific input x in D1. Hence, [a] is a stronger constraint than [b]. However, in the case where the decoder d2 does not properly maps to the target domain D2  (e.g., at the beginning of training, the generated faces are not realistic until the GAN kicks in), then [a] does not bring any information about embeddings from real D2 samples, contrary to [b].
>
> Comparison to baselines
> ---------------------------------
> The main differences with the UNIT paper [b] is we impose stronger constraints on the learned embeddings, i.e. we make use of (i) the domain-adversarial loss [c] and (ii) the semantic consistency loss (rather than pixel-level cycle consistency) to constrain the learned embedding explicitly, while UNIT only relies on weight sharing in the encoder. We will include the COGAN reference in future revisions.
>
>
> Experimental validation
> -------------------------------
> We did not thoroughly investigate previous pixel-level translation tasks as they were not our main focus, but we agree that additional experiments would definitely support the proposed model.
> We also experimented on SVHN to MNIST to compare to the DTN baseline, however we did not observe significant improvement in the classification accuracy compared to these baselines. We omitted these results as they did not seem significant enough for a task like MNIST classification.

---

### Official Review · AnonReviewer1 · 2017-11-27
**The new task and new dataset are nice contributions. Technical novelty is limited. Experiment design could be improved.**

**Rating:** 4
**Confidence:** 4

**Review:**

This paper proposed an X-shaped GAN for the so called semantic style transfer task, in which the goal is to transfer the style of an image from one domain to another without altering the semantic content of the image. Here, a domain is collectively defined by the images of the same style, e.g., cartoon faces.

The cost function used to train the network consists of five terms of which four are pretty standard: a reconstruction loss, two regular GAN-type losses, and an imitation loss. The fifth term, called the semantic consistency loss, is one of the main contributions of this paper. This loss ensures that the translated images should be encoded into about the same location as the embedding of the original image, albeit by different encoders.

Strengths:
1. The new CartoonSet dataset is carefully designed and compiled. It could facilitate the future research on style transfer.
2. The paper is very well written. I enjoyed reading the paper. The text is concise and also clear enough and the figures are illustrative.
3. The semantic consistency loss is reasonable, but I do not think this is significantly novel.

Weaknesses:
1. Although “the key aim of XGAN is to learn a joint meaningful and semantically consistent embedding”, the experiments are actually devoted to the qualitative style transfer only. A possible experiment design for evaluating “the key aim of XGAN” may be the facial attribute prediction. The CartoonSet contains attribute labels but the authors may need collect such labels for the VGG-face set.
2. Only one baseline is considered in the style transfer experiments. Both CycleGAN and UNIT are very competitive methods and would be better be included in the comparison.
3. The “many-to-many” is ambiguous. Style transfer in general is not a one-to-one or many-to-one mapping. It is not necessary to stress the many-to-many property of the proposed new task, i.e., semantic style transfer.

The CartoonSet dataset and the new task, which is called semantic style transfer between two domains, are nice contributions of this paper. In terms of technical contributions, it is not significant to have the X-shaped GAN or the straightforward semantic consistency loss. The experiments are somehow mismatched with the claimed aim of the paper.

---

> ### Author Response · Authors · 2018-01-05
> **Response to reviews**
>
> We thank the reviewers for their comments, in particular for their suggestion of quantitative evaluation.
>
> (Response to Weaknesses 2/) We originally focused on the DTN work as it was the closest in terms of motivation and applications. As for more recent work, the CycleGAN paper mentions that the method often fails on task where input and output are significantly different in structure (e.g., cat to dog) so it might be a weak baseline for face-to-cartoon. However, as was also mentioned by the other reviewers,  we agree that UNIT could act as another strong baseline for the task as it seems to allow feature-level transfer between the two domains.

---

> ### Comment · AnonReviewer1 · 2018-01-17
> **Post rebuttal**
>
> The rebuttal is brief and does not address my major concerns. To improve the paper, the authors may consider to include more baselines and some ablation studies of the proposed method. Additionally, the clarify and presentation of the paper could be improved too.

---

### Official Review · AnonReviewer2 · 2017-12-03
**While the results look decent and the method appears sound, ultimately this paper does not have convincing experiments and does not contribute a clear advance over prior work.**

**Rating:** 4
**Confidence:** 3

**Review:**

This paper proposes a new GAN-based model for unpaired image-to-image translation. The model is very similar to DTN [Taigman et al. 2016] except with trained encoders and a domain confusion loss to encourage the encoders to map source and target domains to a shared embedding. Additionally, an optional teacher network is introduced, but this feels rather tangential and problem-specific. The paper is clearly presented and I enjoyed the aesthetics of the figures. The method appears technically sound, albeit a bit complicated. The new cartoon dataset is also a nice contribution.

My main criticism of this paper is the experiments. At the end of reading, I don’t know clearly which aspects of the method are important, why they are important, and how the proposed system compares against past work. First, the baselines are insufficient. Only DTNs are compared against, yet there are many other recent methods for unpaired image-to-image translation, notably, cycle-consistency-based methods and UNIT. These methods should also be compared against, as there is little evidence that DTNs are actually SOTA on cartoons (rather, the cartoon dataset was not public so other papers did not compare on that dataset). Second, although I appreciated the ablation experiments, they are not comprehensive, as discussed more below. Third, there is no quantitative evaluation. The paper states that quantifying performance on style transfer is an unsolved problem, but this is no excuse for not at least trying. Indeed, there are many proposed metrics in the literature for quantifying style transfer / image generation, including the Inception score [Salimans et al. 2016], conditional variants like the FCN-score [Isola et al. 2017], and human judgments. These metrics could all be adapted to the present task (with appropriate modifications, e.g., switching from Inception to a face attribute classifier). Additionally, as the paper mentions at the end, the method could be applied to domain adaptation, where plenty of standard metrics and benchmarks exist.

Ultimately, the qualitative results in the paper are not convincing to me. It’s hard to see the advantages/disadvantages in each comparison. For example in Figure 8, it’s hard to even see any overall change in the outputs due to ablating the semantic consistency loss and the teacher loss (especially since I’m comparing these to Figure 6, which is referred to “Selected results” and therefore might not be a fair comparison). Perhaps the effect of the ablations would be clearer if the figures showed a single input followed by a series of outputs for that same input, each with a different term ablated. A careful reader might be able to examine the images for a long time and find some insights, but it would be much better if the paper distilled these insights into a more concise and convincing form. I feel sort of like I’m looking at raw data, and it still needs to be analyzed.

I also think the ablations are not sufficiently comprehensive. In particular, there is no ablation of the domain adversarial loss. This seems like an important one to test since it’s one of the main differences from DTNs. I was a bit confused by the “finetuned DTN” in Section 7.2. Is this an ablation experiment where the domain adversarial loss and teacher loss are removed? If so, referring to it as so may be clearer than calling it a finetuned DTN. Interestingly, the results of this method look pretty decent, suggesting that the domain adversarial loss might not be having a big effect, in which case XGAN looks very close indeed to DTNs. It would be great here to actually quantify the mentioned sensitivity to hyperparameters.

In terms of presentation, at several points, the paper argues that previous, pixel-domain methods are more limited than the proposed feature-space method, but little evidence is given to support these claims. For example, “we argue that such a pixel-level constraint is not sufficient in our case” in the intro, and “our proposed semantic consistency loss acts at the feature level, allowing for more flexible transformations” in related work. I would like to see more motivation for these assertions, and ultimately, the limitations should be concretely demonstrated in experiments. In models like CycleGAN the pixel-level constraint is between inputs and reconstructed inputs, and I don’t see why this necessarily is overly restrictive on the kinds of transformations in the outputs. The phrasing in the current paper seems to suggest that the pixel-level constraints are between input and output, which, I agree, would be directly restrictive. The reasoning here should be clarified. Better yet would be to provide empirical evidence that pixel-domain methods are not successful (e.g., by comparing against CycleGAN).

The usage of the term “semantic” is also somewhat confusing. In what sense is the latent space semantic? The paper should clarify exactly what this term refers to, perhaps simply defining it to mean a “low-dimensional shared embedding.”

I think the role of the GAN objective is somewhat underplayed. It is quite interesting that the current model achieves decent results even without the GAN. However, there is no experiment keeping the GAN but ablating other parts of the method. Other papers have shown that a GAN objective plus, e.g., cycle-consistency, can do quite well on this kind of problem. It could be that different terms in the current objective are somewhat redundant, so that you can choose any two or three, let’s say, and get good results. To check this, it would be great to see more comprehensive ablation experiments.


Minor comments:
1. Page 1: I wouldn’t call colorization one-to-one. Even though there is a single ground truth, I would say colorization is one-to-many in the sense that many outputs may be equally probable according to a Bayes optimal observer.
2. Fig 1: It should be clarified that the left example is not a result of the method. At a glance this looks like an exciting new result and I think that could mislead casual readers.
3. Fig 1 caption: “an other” —> “another”
4. Page 2: “Recent work … fail for more general transformations” — DiscoGAN (Kim et al. 2017) showed some success beyond pixel-aligned transformations
5. Page 5: “particular,the” —> “particular, the”; quotes around “short beard” are backwards
6. Page 6: “founnd” —> “found”
7. Page 11: what is \mathcal{L}_r? I don’t see it defined above.

---

> ### Author Response · Authors · 2018-01-05
> **Reponse to reviews**
>
> We thank the reviewer for their detailed comments and helpful suggestions. As was raised by other reviewers, we take note of the lack of experimental validation. In the following we address some more specific issues addressed by the reviewer.
>
> Ablation experiments
> ----------------------------
> Indeed, the finetuned DTN would be equivalent to XGAN with a fully shared encoder, only one decoder (so no reconstruction loss on the source domain), and no domain-adversarial nor teacher loss.
>  On the long term, the combination of GAN + semantic consistency loss has a similar effect to the domain-adversarial loss; however including the domain-adversarial loss  should lead faster to a  regime where the embeddings for both domains lie closer.
>
> Comparison to Baselines
> ----------------------------------
> Our main reasons for not including CycleGAN as a baseline were (i) the original paper claims in the conclusion that experiments on translation between significantly different domains were unsuccessful and (ii) CycleGAN uses fully convolutional networks (no latent representation bottleneck) hence we hypothesize it strongly retains local pixel information, even though there is no explicit pixel-level constraint between input and output.
> We agree that UNIT would be a well fitted baseline to compare to on the face-to-cartoon task.

---

> > ### Comment · AnonReviewer2 · 2018-01-13
> > **Response to rebuttal**
> >
> > I thank the authors for their response to my review. However, I stand by my initial assessment. I think this paper suggests an interesting model but it needs much more extensive experimentation to prove its utility, especially since rather similar alternatives already exist in the literature. I'm glad to see the authors think UNIT would be a good baseline to compare against, and I would encourage them to also try CycleGAN/DiscoGAN/DualGAN. I expect those methods would do well despite the inductive bias toward pxiel-level correspondence. Even if they perform poorly, that should be tested and not simply asserted.

---

### Public Comment · ~R_Devon_Hjelm1 · 2017-11-08
**Many to many?**

It's unclear how this model is many-to-many. The mappings are deterministic as far as I can tell, no?

---

> ### Author Response · Authors · 2017-11-14
> **Clarification on 'many-to-many'**
>
> Thank you for the question. To clarify, the *tasks* we consider are many-to-many in the sense there is no pre-defined one-to-one mapping between the domains, eg one face maps to many possible cartoons and vice-versa. However, the face/cartoon-to-latent-space mapping is many-to-one. We indeed only report results from deterministic models, which can be trivially extended to be conditional to a noise vector as well. An example of how this can be done is outlined in the CVPR 2017 PixelDA paper: Unsupervised Pixel-level Domain Adaptation with GANs by Bousmalis et al. In this and other papers, they found that introducing such noise does not affect the quality of the generated samples.

---

> > ### Public Comment · ~R_Devon_Hjelm1 · 2017-11-27
> > **many to many**
> >
> > I disagree that extending to conditioning on noise is "trivial", as this is a well-known alignment problem in unsupervised domain mapping. Please see
> > https://arxiv.org/pdf/1709.00074.pdf

---

### Public Comment · (anonymous) · 2017-12-14
**Publish the code and datasets**

could the code and data sets  be made public?

---

> ### Author Response · Authors · 2018-01-05
> **Dataset publication**
>
> The dataset is in the process of being made public. We will update the submission as soon as it is available.

---

### Public Comment · (anonymous) · 2018-01-23
**How is the semantic consistency loss backpropagated?**

The semantic consistency loss is ||e2(d2(e1(x1))), e1(x1)||. There are two possible implementations:
1. Treat e1(x1) as the label/target and do not back-propagate through it.
2. Use the gradient w.r.t. the e1(x1) to update e1. So the e1 is updated with two gradient flows.

I'm wondering which one the authors use.

---

### Decision · Program_Chairs · 2018-01-29
**ICLR 2018 Conference Acceptance Decision**

**Decision:**

Reject

**Comment:**

This paper was reviewed by 3 expert reviewers. All three recommend rejection citing significant concerns (e.g. missing baselines).